# A Study of the Susceptibility of Laboratory Animals to the Lumpy Skin Disease Virus

**DOI:** 10.3390/life13071489

**Published:** 2023-06-30

**Authors:** Elena Yurievna Pivova, Mikhail Evgenievich Vlasov, Timofey Aleksandrovich Sevskikh, Olga Sergeevna Povolyaeva, Sergey Petrovich Zhivoderov

**Affiliations:** Federal Research Center for Virology and Microbiology (FRCVM), 601125 Volginsky, Russia; vlasovmikhail1993@yandex.ru (M.E.V.); 2741188@mail.ru (O.S.P.); zhivoderov-serg@mail.ru (S.P.Z.)

**Keywords:** lumpy skin disease in cattle, ELISA, real-time PCR, cell culture, laboratory animals

## Abstract

This article presents the results of a study on the susceptibility of laboratory animals to the lumpy skin disease virus (LSDV). Mice weighing 15–20 g, hamsters weighing 40–60 g, guinea pigs weighing 600–1200 g, and rabbits weighing 2.5–3 kg were used in this study. Nodules were observed on the skin of rabbits and hamsters at the sites of inoculation. The virus was isolated from the affected skin areas in cell culture and examined using real-time PCR, indicating its tropism for animal skin. The production of anticapripoxvirus antibodies was detected using the neutralization reaction, starting from 10 days after infection in mice, 27 days in rabbits, and 14 days in hamsters. Some laboratory animals exhibited multiple skin nodules. This indicates that these animal species may play a role in maintaining the epizootic process.

## 1. Introduction

Lumpy skin disease (LSD) is a contagious viral disease that spreads through vectors and is characterized by persistent fever; skin nodules; edema of subcutaneous tissue and organs; and damage to the lymphatic system, eyes, and mucous membranes of respiratory and digestive organs [1]. The causative agent of the disease is an enveloped DNA-containing virus in the Capripoxvirus genus of the Poxviridae family, which shares antigenic characteristics with sheep pox and goat pox viruses [2].

The disease can be transmitted by sick animals and asymptomatic carriers, with mechanical vector-borne spread being the most common. Studies have shown that Aedes aegypti mosquitoes can transmit the virus within 2–6 days after the pathogen enters the insect’s body with the blood of infected animals [3]. Ixodic ticks of the *Rhipicephalus*, *Amblyomma*, and *Hyalomma* genera found in South Africa can also mechanically transmit the virus [4,5]. Outbreaks of LSD have been linked to the activity of the autumn beetle fly (*Stomoxys calcitrans*) [6]. The contact route of transmission has not been proven, as intact animals that were in the same box with infected animals for a month remained healthy [7].

Cattle are susceptible to the LSDV, while other animals such as giraffes and impalas have also shown signs of the disease in experimental conditions [8]. Laboratory animals’ sensitivity to the LSDV varies, with rabbits only showing clinical signs of the disease at the first passages without further manifestations [9,10]. When infected with the LSDV, 1.5-month-old lambs and goats developed nodules on the skin at injection sites, with the diameter increasing from 2 to 5 cm over seven days. The viral genome was detected in blood and oral swabs, as well as in internal organs (lungs and lymph nodes) [11].

Simultaneously, our studies have shown that the transplanted epithelial cell cultures of elk embryo (EE) and rabbit kidney (RK-13) are sensitive to the LSDV, which accumulates in these cultures up to 3.5–4.5 lg TCID50/cm^3^ [12].

Our research evaluated the susceptibility of laboratory animals such as mice, hamsters, guinea pigs, and rabbits to the LSDV and the possibility of using them as an experimental model for the disease in vivo.

## 2. Materials and Methods

### 2.1. Strain

In our experiments, we utilized the LSDV strain Volgograd, which was deposited in the State Collection of Microorganisms of FRCVM under the number 3161. We also used cultural material from the 15th passage in the transplanted sheep kidney cell line (PO (VNIIVViM)), which had an infectious activity of 5.5 lg TCID50/mL.

### 2.2. Animals

The laboratory animals used in this study included mice, guinea pigs, rabbits, and Syrian hamsters, all of which were obtained from the experimental animal sector of FRCVM and were clinically healthy.

Outbred mice with a live weight of 15–20 g were infected intradermally at a dose of 1.8 × 10^5^ TCID50 at two points with 50 µL and at two points subcutaneously with 250 µL (32 mice) and intranasally at a dose of 0.6 × 10^5^ TCID50 with 200 µL (10 mice). By decapitation, where mice where previously euthanized with alcohol–ether, blood and serum samples were taken.

Four guinea pigs with a live weight of 500–600 g were infected intradermally at a dose of 3.16 × 10^5^ TCID50 at 4 points with 250 µL. Blood and serum samples were collected by heart puncture.

Four rabbits with a live weight of 2.5–3.0 kg were infected intradermally at 2 points with 250 µL and intravenously in the auricle with 500 µL at a dose of 3.16 × 10^5^ TCID50. Blood and serum samples were taken from the marginal ear vein.

Four Syrian hamsters with a live weight of 40–60 g were infected intradermally at 4 points with 250 µL at a dose of 3.16 × 10^5^ TCID50. Blood and serum samples were taken from the posterior vein of the lower leg.

For each animal species, there was a control group which was injected with virus-containing material which was inactivated by heat treatment (65 °C for 30 min).

All animals were kept in microbiological safe boxes and were fed a full balanced diet with plenty of water. The animals were examined daily for clinical signs, including the development of skin nodules at the site of inoculation and in various places on the body.

### 2.3. Sample Collection

Blood and serum samples were collected starting from 7 to 28 days after infection. For blood samples, we used tubes with an anticoagulant (EDTA); for serum samples, a clotting activator was used. The collected samples were stored at a temperature of 4.0 ± 2.0 °C for 24 h. Then, the contents of the test tubes were poured into aliquots (0.1–0.2 mL) in separate sterile test tubes with a volume of 1.5 mL and stored at a temperature of minus 70 ± 0.5 °C until the tests were carried out.

The animals were humanely slaughtered using carbon dioxide (CO_2_), and pathoanatomical autopsies were performed according to the generally accepted method with a description of the state of all organs and systems of the body. To isolate the virus and its genome, we selected the lungs, spleen, and liver from each animal. A 10% suspension was prepared from each organ, and presence of the LSDV genome was examined using real-time PCR.

### 2.4. Real-Time PCR

Nucleic acids were isolated using a set of “RIBOSORB” (LLC “InterLabService”, Moscow, Russia) in accordance with the manufacturer’s instructions. Viral genomic DNA was detected using the Bowden et al. method (16) with the oligonucleotide primers CaPv 074 F1 (5/-AAA ACG GTA TAT GGA ATA GAG TTG GAA-3/), CaPv 074 R1 (5′-AAA TGA AAC CAA TGG ATG GGA TA-3/), and the hybridization probe CaPv-074P1 (5/-6FAM-TGG CTC ATA GAT TTCCT-MGB-NFQ-3/). The reaction mixture included 10 pmol of each primer, 5 pmol of fluorescent probe (CJSC “Syntol”, Moscow, Russia), 2.5 µL 10× of DNA buffer (Thermo Scientific, Waltham, MA, USA), 10 mmol of dNTPs mixture (Thermo Scientific, Waltham, MA, USA), 12.5 mmol of magnesium chloride (CJSC “Syntol”, Moscow, Russia), and 1.5 units activity of recombinant Taq DNA polymerase (Thermo Fisher Scientific, Waltham, MA, USA. Real-time PCR was performed on a detection thermal cycler “DT prime” (LLC “DNA-Technology”, Moscow, Russia) according to the following program: preliminary denaturation for 10 min at 95 °C and 45 amplification cycles (15 s at 95 °C, 1 min at 60 °C). The results were interpreted based on the presence/absence of the intersection of the fluorescence curve with the threshold line automatically set by the device. The result was considered positive if the obtained Ct value did not exceed 40.

### 2.5. Isolation of the Virus from Clinical Samples

Virus isolation was performed using the biopsies of the skin nodules. Obtained samples were used to produce a 10% suspension in Eagle’s medium (Merck KgaA, Darmstadt, Germany) with the addition of antibiotics (penicillin and streptomycin, 200–1000 U per 1 mL, nystatin 20 U per 1 mL). After clarification using centrifugation at 2000 rpm, the suspension was introduced into culture flasks with a confluent monolayer of PO cells (FRCVM). One hour after adsorption, the suspension was removed and a maintenance medium containing 2% bovine serum was added. The suspension was then incubated for 5–6 days at 37 ± 0.5 °C. To determine CTA, vials with cell culture were examined under an Olympus CKX31 inverted microscope (Olympus Co., Shinjuku City, Tokyo, Japan). Vials with cell culture were frozen and stored at minus 40 ± 0.5 °C, then the culture liquid was thawed at room temperature and the next passage was carried out by adding 1 cm^3^ of culture liquid to the cell monolayer. The infectious activity of the virus was determined using titration in a 1–2-day culture of a continuous cell line PO, grown in 96-well microplates. The virus titer was calculated using the method by Reed and Mench [13] and expressed in lg TCID50/cm^3^.

### 2.6. Virus Neutralization Assay

The analysis was carried out using control normal (negative) serum and standardized specific hyperimmune serum against sheep pox in a 1:16 dilution. Samples of the sera under study were inactivated at 56 °C for 30 min and the reaction was set in 2 variants: (1) To determine the titer of virus-neutralizing antibodies in the sera under study, a constant dose of the virus (100–200 TCID50) was added to the double dilutions of the sera; (2) To identify the virus and determine the neutralization index (NI, the same dose of serum was added to tenfold dilutions of the virus in a working dilution of 1:10.

Culture plates were incubated at a temperature of 37 ± 0.5 °C in an atmosphere of 95% humidity and 5% CO_2_ for 6 days. The development of CPA was evaluated by viewing the culture plates using a microscope. The reaction was considered positive if the neutralization index was at least 1.5 [6].

### 2.7. Enzyme-Linked Immunosorbent Assay (ELISA)

To determine the presence of antibodies against the LSDV, the ELISA was carried out with a commercial diagnostic kit IDVET, ID Screen^®^ Capripox Double Antigen Multi-species according to the manufacturer’s instructions.

## 3. Results

### 3.1. Detection of the LSDV Genome and Antibodies

In the experimental infection of mice, no clinical signs were observed. The LSDV genome was detected between 7 and 14 days in 75% of mice after intradermal and subcutaneous infection (Table 1). When infected intranasally, the virus genome was detected in 50% of mice from 7 to 20 days (Table 1).

The results presented in Table 1 show that on day 7, the LSDV genome was detected in 50% of blood samples from white mice, 75% of rabbits, and 100% of guinea pigs and hamsters. In blood samples taken on day 14, the virus genome was detected in 75% of guinea pigs and white mice with intradermal and subcutaneous infection, as well as in 100% of rabbits, hamsters, and intranasally infected white mice.

The results were also confirmed using an ELISA assay that showed the presence of the antibodies against the LSDV. In animals infected intradermally and subcutaneously, the titer of virus-neutralizing antibodies in blood serum on day 14 was 1:8; on days 21 and 28 was 1:4, in those infected intranasally on days 14 and 21 was 1:16, and on day 28 was 1:8 (Table 2).

As shown in Table 2, virus-neutralizing antibodies were detected in infected animals (except for rabbits) from 14 days in titers of 1:4–1:32, with the highest in hamsters. On days 21 and 28 the titer was 1:8–1:16. Antibodies to the LSDV in rabbits were detected in a titer of 1:8 only on day 28 (the observation period).

### 3.2. Results of the Infection and Autopsy of Mice

At the pathoanatomical autopsy of the infected and control group of animals (mice), visible lesions of internal organs were not observed, and the examination of the liver, lungs, and spleen by real-time PCR did not reveal the LSDV genome.

### 3.3. Results of the Infection and Autopsy of Guinea Pigs

After the inoculation of guinea pigs, no clinical signs were observed. However, on day 7, the LSDV genome was detected in the blood of all animals that were infected, and on days 14, 21, and 28, in 75% of the pigs (Table 1). At autopsy, pathoanatomic changes were found in the lungs, which were blood-filled with hemorrhages on the incision (Figure 1). A 10% suspension was prepared from the liver, lungs, and spleen and this was examined for the presence of the genome using real-time PCR. Virus DNA was detected in 100% of lungs, and in 50% of liver and spleen samples.

The DNA of the LSDV was not detected in the blood and the internal organs of the control group of animals (guinea pigs); moreover, there were no detectable antibodies.

### 3.4. Results of the Infection and Autopsy of Rabbits

In infected rabbits, on day 4, a formation of skin nodules with a diameter of 1.0–1.5 cm was noted at the site of inoculation (Figure 2). Nodules on the skin persisted for 4–6 days.

The virus genome was detected in the 10% suspension prepared from nodules and was subsequently isolated in the PO cell culture, where the infectious activity was equal to 2.5 lg TCID50/mL. The presence of the genome in the culture suspension was confirmed using qPCR. The identification of the virus was confirmed using the neutralization reaction with a specific serum against sheep pox and the ELISA method. The neutralization index was 2.5.

On the 7th, 21st, and 28th day after infection with the virus, LSD was detected in 75% of animal blood samples using real-time PCR, and on the 14th day it was detected in all animals (Table 1). Virus-neutralizing antibodies appeared on the 28th day; the titer was 1:8 (Table 2). Similar results were obtained using the ELISA.

At the pathoanatomical autopsy of infected and intact rabbits, there were no visible lesions of internal organs; when examining internal organs (liver, lungs, and spleen) for the presence of the LSDV genome using qPCR, negative results were obtained. When the inactivated material was administered to intact rabbits, no lesions were observed at the site of its administration. On days 6–7, rabbit skin samples were taken and the genome was not detected using real-time PCR.

### 3.5. Results of the Infection and Autopsy of Hamsters

Based on the research conducted by Sergeev A.A. and co-authors [14], which utilized steppe marmots of the Baibak species as model biosystems for orthopoxvirus studies, we were interested in investigating the susceptibility of Syrian hamsters to the LSDV. The LSDV was detected in the blood samples of all infected animals on the 7th, 14th, and 21st day, and in 75% of infected animals on the 28th day (Table 1). Virus-neutralizing (VN) antibodies were not detected in the blood sera of animals on day 7 (≤1:2). However, starting from day 14 to day 21, 75% of experimental animals showed a titer of VN antibodies of 1:8–1:16, and on day 28, 1:4–1:8 (Table 2). Similar results were also obtained through ELISA testing.

In two white Syrian hamsters (albinos), one pustule of size 0.5 cm^3^ was formed on the skin on the 10th day (Figure 3). The virus genome was detected in a 10% suspension prepared from a pustule using real-time PCR. From the same suspension in the transplanted PO cell culture, the LSDV was isolated; its titer was 2.5 lg TCID50/mL. The cytopathic effect of the virus is shown in Figure 4. The presence of the genome in the culture suspension was confirmed using real-time PCR. The identification of the virus was confirmed using the neutralization reaction with a specific serum against sheep pox, as well as the ELISA method. The neutralization index was 2.5. In the prepared suspensions of internal organs (lungs, spleen, and liver), the LSDV genome was not detected.

When examining blood and internal organs from animals of the control group, DNA of the LSDV was not detected using real-time PCR. No antibodies were detected during serological reactions (neutralization reaction, ELISA).

## 4. Discussion

The replication of the lumpy skin disease (LSD) virus in different cell cultures has been demonstrated by both foreign and domestic researchers [12,15,16]. Lamb, goat, and calf organs are used to derive primary and continuous cell cultures for the isolation and propagation of the LSD virus. Adhering to the cultivation scheme is crucial for the successful isolation and identification of the virus. Continuous cell lines show promise for LSD-virus cultivation as they allow the production of virus-containing material in large quantities. This material derived from continuous cell lines is widely used to study the biological and molecular–genetic characteristics of the virus.

However, there is limited research on the isolation methods, optimization of cultivation parameters, and susceptibility of laboratory animals to the virus. For example, Sergeev et al. established model biosystems (Baybak ground squirrels) for studying orthopoxviruses [14]. As there is currently no laboratory model available to replicate the infectious process, and economically valuable animals are expensive, understanding the susceptibility of laboratory animals to the LSD virus is an important aspect of disease studies and vaccine development.

Based on our research on the susceptibility of various laboratory animals (rabbits, guinea pigs, Syrian hamsters, and outbred white mice) to the LSD virus, rabbits and Syrian hamsters (albinos) may be suitable laboratory models. These animals developed cutaneous nodules at the site of inoculation upon infection, which resolved within 8–10 days. Rabbits showed clinical signs of the disease characteristic of LSD (skin nodules) on the 4th day after infection, while albino Syrian hamsters showed these signs on the 10th day. The viral genome was detected in the blood of infected animals 7 days post-infection, and virus-neutralizing antibodies began to be detectable from 14 days post-infection.

The data obtained from infecting albino rabbits and Syrian hamsters show that the LSDV reproduces in the bodies of these animals, with the manifestation of clinical signs and the formation of an immune response. This indicates that these animal species may play a role in maintaining the epizootic process. Our results show that Syrian hamsters are the most susceptible species for the LSDV virus and, thus, could be used as an in vivo model for LSDV infection. However, additional studies are needed to investigate the ability of the LSDV to be isolated from the secretions and excreta of infected laboratory animals.

## Figures and Tables

**Figure 1 life-13-01489-f001:**
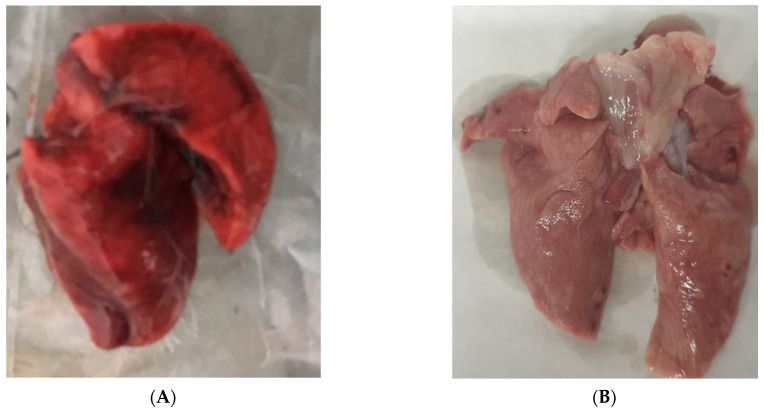
(**A**) Affected guinea pig lungs. (**B**) Lungs of an intact guinea pig.

**Figure 2 life-13-01489-f002:**
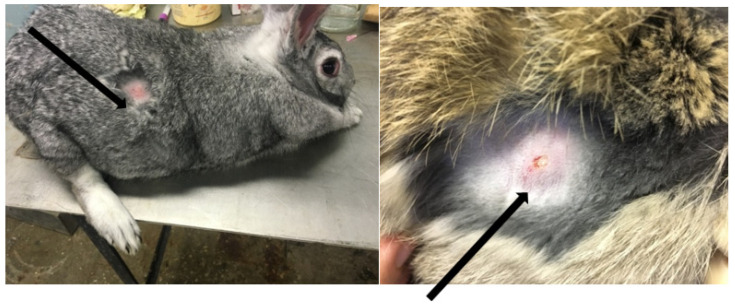
A nodule on the skin of a rabbit infected with the LSDV.

**Figure 3 life-13-01489-f003:**
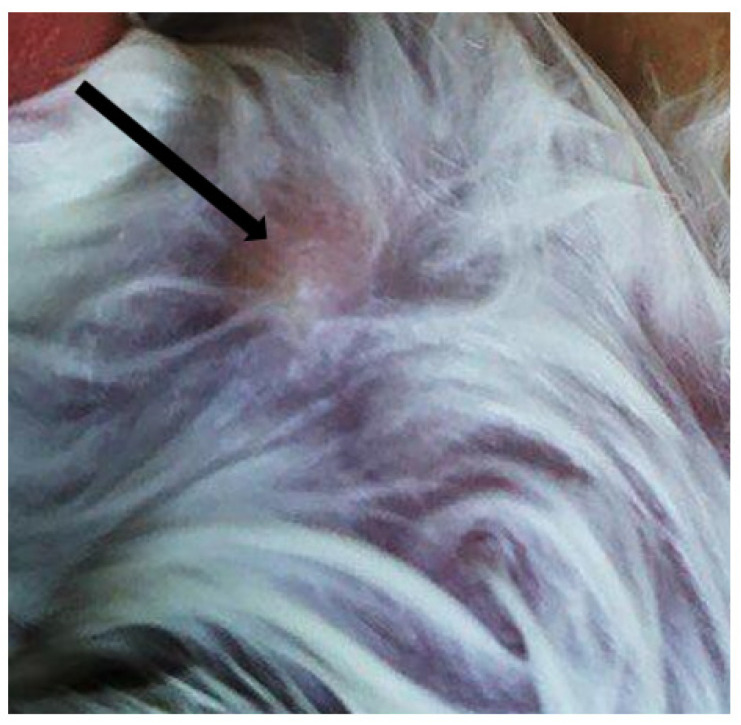
Nodule in albino hamster.

**Figure 4 life-13-01489-f004:**
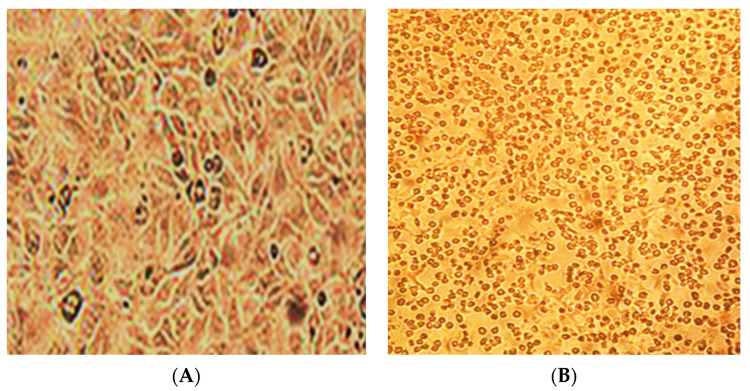
Cytopathic effect of the LSDV in the continuous PO cell line (**A**,**B**). (**A**)—control cell culture, (**B**)—on the 5th day of infection. Light microscopy. (Magnification ×150).

**Table 1 life-13-01489-t001:** Detection of the LSDV genome in blood samples of infected laboratory animals.

Animal	Type of Administration	The Results of the Detection of the LSDV Genome (Days)			
		7	14	21	28
Rabbits	i/d, i/v	75%	100%	75%	75%
Guinea pigs	i/d	100%	75%	75%	75%
Hamsters		100%	100%	100%	75%
White mice	i/d, s/c	50%	75%	0%	0%
	i/n	50%	100%	50%	0%

Note: i/d—intradermally, s/c—subcutaneous, i/n—intranasally, i/v—intravenous; «%»—results in animal group.

**Table 2 life-13-01489-t002:** Detection of virus-neutralizing antibodies in laboratory animals infected with the LSDV.

Animal	Type of Administration	Titer of Antibodies to the LSDV in Blood Serum Samples (Days)			
		7	14	21	28
Rabbits	i/d, i/v	-	-	-	1:8
Guinea pigs	i/d	-	1:4	1:8–1:16	1:16
Hamsters		-	1:8–1:32	1:8–1:16	1:4–1:8
White mice	i/d, s/c	-	1:8	1:4	1:8
	i/n	-	1:16	1:16	1:8

Note: i/d—intradermally, s/c—subcutaneous, i/n—intranasally, i/v—intravenous.

## Data Availability

The data presented in this study are available on request from the corresponding author.

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
