# Peer review of "A Study of the Susceptibility of Laboratory Animals to the Lumpy Skin Disease Virus"

_life, 2023, doi:10.3390/life13071489_

Round 1

Reviewer 1 Report

In 2017, researchers did their very similar study as in vitro.

It can be seen in doi: 10.15389/agrobiology.2017.6.1265rus, doi:10.15389/agrobiology.2017.6.1265eng.

I also recommend reference to the mentioned literature of Balysheva. Additionally, I have attached Turnitin report of this study. 

In mentioned study, some cell lines originated from monkey, porcine, elk, sheep, calf, etc. have been used for Lumpy Skin Disease Virus propagation.

Also they have done in vivo study that similar with in vitro study currently. But, this study on the susceptibility of laboratory animals to the Lumpy Skin Disease Virus it may be more useful if it can be converted into the attenuation of a vaccine strain.

Author Response

Response to reviewer #1.

The research you mentioned in review was authored by same researchers as this study. It was a part of a big research project concerning transmission of LSDV. We did not cite it because self-citation is not approved in Russia so much, but we will add this reference in manuscript. 

Thank you very much for evaluation of our work! We believe that this data will be useful for risk assesment and probably for future vaccine research.

Reviewer 2 Report

The article is about the susceptibility of laboratory animals such as mice, hamsters, guinea pigs, and rabbits to the Lumpy skin disease virus.

Material and methods:

The ethics statement is not provided in the methods.

Line 55-91: The authors may write 1st part of the methods with the headings, such as LSD virus strain used in the study, virus inoculation in lab animals, lab conditions and sample collection, etc.

Line 91: the genome was examined by RT-PCR???????. Please explain or correct the sentence. Did you perform any genome sequencing?

Line 136-139: Please provide a brief procedure to determine the presence of antibodies through ELISA.

There is no reference in methodology. Please provide references in all sections. 

Results and discussion: 

The authors must improve results and discussions.

The authors did not discuss the results with the findings of others. This is not the 1st study across the globe. My apologies.

The experiment is good but the writing and presentation of results need substantial improvement. There is no data analysis.

The authors should write/re-write the discussion. The discussion is missing.

Line 145: Table 1. Detection of LSDV genome????

Line 199: Where is the reference?

Moderate editing of the English language is required.

Author Response

Response to reviewer #2.

Point #1. The ethics statement is not provided in the methods.

Response to point #1. We provided ethics statement in "Institutional Review Board Statement" (lines 245-247) according to requirements of the journal.

Point #2. Line 55-91: The authors may write 1st part of the methods with the headings, such as LSD virus strain used in the study, virus inoculation in lab animals, lab conditions and sample collection, etc.

Response to point #2. Thank you for this point, we will divide this section with the headings.

Point #3. Line 91: the genome was examined by RT-PCR???????. Please explain or correct the sentence. Did you perform any genome sequencing?

Response to point #3. Yes, we meant the detection of viral genome. We corrected it in revised version of manuscript.

Point #4. Line 136-139: Please provide a brief procedure to determine the presence of antibodies through ELISA.

There is no reference in methodology. Please provide references in all sections. 

Response to point #4. We carried out ELISA procedure according to manufacterer's instructions. 
We added references in M&M section, thank you for this point.

Point #5. Results and discussion: 

The authors must improve results and discussions.

The authors did not discuss the results with the findings of others. This is not the 1st study across the globe. My apologies.

The experiment is good but the writing and presentation of results need substantial improvement. There is no data analysis.

The authors should write/re-write the discussion. The discussion is missing.

Response to point #5. We referenced the only similar study on rabbits. We could not find any similar data on LSDV infection in laboratory animals. We believe that our study is not the first across the globe, but unfortunately we could not prove our beliefs.

That is why we chose the format of brief report. This is only dry summary of our results. And due to the lack of similar results, there is not much to discuss.

Point #6. Line 145: Table 1. Detection of LSDV genome????

Response to point #6. Yes...

Point #7. Line 199: Where is the reference?

Response to point #7. Thank you for pointing this out. We added the correct reference in revised document.

Reviewer 3 Report

The aim of this study was to determine the susceptibility of laboratory animals to the Lumpy Skin Disease Virus that is quite interesting for the readers. However, there are some points need to add and verify to make it clear and easy to understand.

Abstract line 20-22 it seems to be high light of the work but when look thru the manuscript, the conclusions was the high light of the work. Therefore, it should conclude in the same way.

Introduction - need to add more review on laboratory animal and LSD.

Results - it might be nice to have one more table on PCR result, what organs that were positive and which animals and how long to detect.

On line 189, 191 27th day it should be 28th day.

Discussion – line 207 in 2 white Syrian hamsters (albinos) found the skin lesion. That’s interesting. On the outbreak last year, I observed that the brown color beef cows has severe clinical signs than white color beef cows and Beef cows has more severe clinical sign than dairy cows. On the difference between breed can point out that may come from health status and management, but for the color, it was still suspicious. Are you have any idea on this?

Author Response

Response to reviewer #3.

Point #1. Abstract line 20-22 it seems to be high light of the work but when look thru the manuscript, the conclusions was the high light of the work. Therefore, it should conclude in the same way.

Response to point #1. Thank you for this comment, we agree with this and we changed an abstract in revised manuscript to correctly highlight our results.

Point #2. Introduction - need to add more review on laboratory animal and LSD.

Response to point #2. We would like to add more review on LSDV in laboratory animals, but unfortunately we did not manage to find any studies or data on this matter. That was the reason why we began study this question. The only piece of information that we managed to find was concerning LSDV-infected rabbits in EFSA's scientific opinion https://efsa.onlinelibrary.wiley.com/doi/pdf/10.2903/j.efsa.2015.3986. " Generalised skin lesions can be detected in LSDV-infected rabbits." - and that's it (as you can see, it is already in the list of references) Other studies were focused on cell cultures, not exactly the possibility of LSD-infection in lab animals. 

Point#3. Results - it might be nice to have one more table on PCR result, what organs that were positive and which animals and how long to detect.

Response to point#3. We detected LDSV genome only in guinea pigs' organs and only after their slaughter on 28-th day. So there is not enough information for additional table. 

Point#4 On line 189, 191 27th day it should be 28th day.

Response to point#4. Thank you very much for noting that, it was clearly a missprint that we did not notice. We corrected it in revised document.

Point #5. On the difference between breed can point out that may come from health status and management, but for the color, it was still suspicious. Are you have any idea on this?

Response on point #5. This is very interesting. We never encountered such an issue in our practice. Maybe it has something to do with production of pigments and their influence on some physiological processes. It is well known that albino people have weaker immunity than average human, it was connected to some proteins that play role in both melanocytes and immune cells development (DOI: 10.2174/1566524023362258). We think it can be very interesting theme for future studies.

Reviewer 4 Report

The manuscript entitled “A study of the susceptibility of laboratory animals to the 2 Lumpy skin disease virus” by Pivova et al., is a brief report that evaluated the susceptibility of laboratory animals such as mice, hamsters, guinea pigs, and 14 rabbits to LSDV and the possibility of using them as an experimental model for the disease in vivo. It’s very interesting and useful content. However, yet need more clarification on the results and discussion.

1.      Why you have mentioned “RT-PCR”. LSDV is a DNA virus, so no reverse transcriptase (RT) is required. If you are meaning it’s a real-time PCR (qPCR) then please write qPCR instead of RT-PCR.

2.      On page 141: „no clinical signs were observed in mice.” Have you measured body temperature? As you find a positive LSDV genome in blood samples that means there was viremia and fever should be observed.

3.      On page 164: “examination of the liver, lungs, and spleen by RT-PCR did not reveal the LSDV genome” and on page 171: “Virus DNA was detected in the 100% of lungs and in 50% of liver and spleen samples” and again on page 194: “when examining internal organs (liver, lungs, spleen) for the presence of the LSDV genome by RT-PCR negative results were obtained” is not contradictory to each other? LSD virus mostly localizes in the skin tissue with visible nodules. Have you taken a sample from the skin nodule and tested it by PCR? It would be great to have these results. That is most important to test rather than the internal organs.

4.      Table 1: what were the CT value ranges for each PCR? Could be useful to include it in the table as will give more clarity to the result.

5.      Figure 1A: Affected guineapig lungs: picture is not clear. How can you be confirmed that those lungs change due to LSD as LSDV was not detected in the lung tissue by qPCR? It may be due to a secondary bacterial infection. Can you discuss a little bit, about why you are expecting lesions in internal organs with the virus of genus Poxviridae? I would delete Figure 1. It might not be relevant.

6.      Figure 2: Nice picture. Have you done a qPCR from the tissue taken from this nodule? If not, you should. It will give you clarity on the local skin urticaria/trash, due to injection or if it’s a true skin nodule due to LSDV infection.

7.      Figure 4A: picture is not clear. Please provide high resolution.  Did you test cell culture suspension by qPCR against LSDV? What are the results?

8.      Rabbit and hamster showed nodular signs. Any other clinical signs like a fever?

Author Response

Response to reviewer #4.

Point #1. Why you have mentioned “RT-PCR”. LSDV is a DNA virus, so no reverse transcriptase (RT) is required. If you are meaning it’s a real-time PCR (qPCR) then please write qPCR instead of RT-PCR.

Response to point#1. Yes indeed, we meant qPCR, RT-PCR was incorrect translation. We revised that in manuscript.

Point #2. On page 141: „no clinical signs were observed in mice.” Have you measured body temperature? As you find a positive LSDV genome in blood samples that means there was viremia and fever should be observed.

Response to point #2. Unfortunately, we did not have the suitable device for temperature measurement in mice.

Point #3.   On page 164: “examination of the liver, lungs, and spleen by RT-PCR did not reveal the LSDV genome” and on page 171: “Virus DNA was detected in the 100% of lungs and in 50% of liver and spleen samples” and again on page 194: “when examining internal organs (liver, lungs, spleen) for the presence of the LSDV genome by RT-PCR negative results were obtained” is not contradictory to each other? LSD virus mostly localizes in the skin tissue with visible nodules. Have you taken a sample from the skin nodule and tested it by PCR? It would be great to have these results. That is most important to test rather than the internal organs.

Response to point #3. We don't see any contradictory here. We did not detect LSDV in internal organs of mice (page 164), rabbits (194) and hamsters (215), but there was presence of viral genome in organs of guinea pigs (page 171).
There was no nodules in guinea pigs and mice, only in hamsters and rabbits. We detected LSDV in both  hamsters' and rabbits' nodules (see pages 209-214 and 181-186).

Point #4. Table 1: what were the CT value ranges for each PCR? Could be useful to include it in the table as will give more clarity to the result.

Response to point #4. Threshold for PCR was 40-th cycle, the average value in positive results were around 24-th cycle for all animals. We will consider including this data in table.

Point #5.   Figure 1A: Affected guineapig lungs: picture is not clear. How can you be confirmed that those lungs change due to LSD as LSDV was not detected in the lung tissue by qPCR? It may be due to a secondary bacterial infection. Can you discuss a little bit, about why you are expecting lesions in internal organs with the virus of genus Poxviridae? I would delete Figure 1. It might not be relevant.

Response to point #5. Unfortunately, we do not have better photos of affected lungs. And we detected virus in lungs (see page 172, "Virus DNA was detected in the 100% of lungs, and in the 50% of liver and spleen samples").

Point #6.   Figure 2: Nice picture. Have you done a qPCR from the tissue taken from this nodule? If not, you should. It will give you clarity on the local skin urticaria/trash, due to injection or if it’s a true skin nodule due to LSDV infection.

Response to point #6. Yes, we did qPCR, see page 181 "The virus genome was detected in the 10% suspension prepared from nodules ".

Point #7.  Figure 4A: picture is not clear. Please provide high resolution.  Did you test cell culture suspension by qPCR against LSDV? What are the results?

Response to point #7. Unfortunately, we do not have this pictures in better resolution. And yes, we tested the cell culture in qPCR. Check page 212 "The presence of the genome in the culture suspension was confirmed by qPCR.".

Point #8. Rabbit and hamster showed nodular signs. Any other clinical signs like a fever?

Response to point #8. No, there were no other clinical signs except for nodules.

Round 2

Reviewer 1 Report

It is good to explain the scientific contribution of the article in the conclusion section. So which animal is recommended as a in vivo model for LSDV infection?

Author Response

Author response:
Thank you very much for your feedback and insightful comments! Based on our results, we consider Syrian hamster as the most suitable laboratory animal model for LSDV. We included sentence about it in the conclusions.

Reviewer 4 Report

The authors have improved the manuscript by more clarification of my previous queries. 

Author Response

Thank you very much for your time and effort! We are grateful for insightful comments and valuable improvements to our article!